electrical engineering/geophysics/geometry

GNSS, drones, UAVs, polar, satellite-based augmentation, scintillation

**Author for correspondence:**
Iain Sheridan
e-mail: iainsheridan@hotmail.com

# Drones and global navigation satellite systems: current evidence from polar scientists

## Iain Sheridan

Scott Polar Research Institute, Cambridge University, Cambridge, UK (2018–2019)

 IS, 0000-0002-9149-7715

Aerial unmanned vehicles, so-called drones, present a paradigm shift away from the long-term use by scientists of manned aeroplanes and helicopters. This is evident from the number of research articles that focus on data obtained with drones. This article examines the use of aerial drones for scientific research in cryospheric regions, especially Antarctica and the Arctic. Specifically, it aims to provide insights into the choices and performance of global navigation satellite systems (GNSS) use for drones, including augmentation systems. Data on drone GNSS navigation and positioning in the context of scientific polar research have been scarce. Drone survey data obtained from polar scientists in April 2019 is the first representative sample from this close-knit global community across the specialisms of climatology, ecology, geology, geomorphology, geophysics and oceanography. The survey results derived from 16 countries revealed that 14.71% of scientists used GALILEO, 27.94% used GLONASS and 45.59% used GPS. Many used a combination of two or more GNSS. Multiple regression analysis showed that there is no strong relationship between a specific pattern of GNSS augmentation and greater positioning accuracy. Further polar drone studies should assess the effects of phase scintillation on all GNSS, therefore BEIDOU, GALILEO, GLONASS and GPS.

## 1. Introduction

Unmanned aerial vehicles and unmanned underwater vehicles (singularly and collectively 'drones') are already a crucial instrument of scientific research. This is evident from the rapid increase in the number of articles that focus on research data obtained with drones. For instance, a whole edition of the *International Journal of Remote Sensing*, comprising of 65 articles, has focused solely on aerial drone research. Shortly after, the same journal published another special edition also dedicated to drone research, comprising of 36 articles [1]. In summary, 101 drone articles in less than 24 months.

Drones are also increasingly assisting research not only at high latitudes [2], but also in all cryospheric regions [3]. In both these

locations, drones support important research on climatology, ecology, geology, geomorphology, geophysics and oceanography.

Whatever the precise research, all scientists need to navigate a drone from a known point of departure to perform specific research tasks. The drone has to then return safely with data or samples or both. In the atmosphere of Earth, aerial drones navigate and verify accurate positioning based on one or more Global Navigation Satellite Systems (GNSS). Furthermore, other satellite systems and diverse augmentation methods refine the accuracy of GNSS measurements. Therefore, a common denominator of drone navigation is to understand how each GNSS operates.

## 2. Overall approach

In April 2019, a global drone survey of scientists was completed (hereafter 'the drone survey'). The target sample of 211 scientists was a balanced international mix: Asian, European, Latin American, North American and Russian institutions. Out of these 211 scientists, 42 responded with detailed answers on their choices of GNSS, GNSS performance and GNSS augmentation accuracy achieved.

## 3. Global navigation satellite systems

Before covering the detail of the drone survey, it is timely to summarize how and why GNSS is relevant to drones used by polar scientists. All aerial drones used at high latitudes and in cryospheric regions rely on GNSS. While not the only basic navigation or data positioning systems available, GNSSs are the primary means of both navigation and positioning for scientists.

GNSS signal availability is determined by three key factors: the altitude of the orbit, the inclination angle of the satellite and the Earth ground-level 'field of view' width of the signal transmitter attached to the satellite.

Each satellite is transmitting two pieces of information: (i) its position in space, and (ii) its clock time. All satellite clocks are synchronized and accurate to one millionth of a second. Applying the equation *distance = speed × time*, the speed of light ($2.99792458 \times 10^8$ m s$^{-1}$) multiplied by the time taken for a satellite signal to arrive at a GNSS receiver, provides the altitude (distance) of the satellite from the user. The example in equation (3.1) (adapted from [4]), based on the known orbit altitude of the US GPS satellites, underlines that each signal reaches a GNSS receiver in approximately a millionth of one second.

$$\left.\begin{aligned} t &= \frac{d}{s}, \\ t &= \frac{26\,600\,(\text{km})}{2.99792458 \times 10^8 \text{ m s}^{-1}}, \\ t &= 26\,600\,000/299\,792\,458, \\ t &= 8.8728 \times 10^{-5} \text{ s} \\ t &= 0.000088728 \text{ s}. \end{aligned}\right\} \tag{3.1}$$

and

The concept of pseudorange is critical to explaining how GNSS works with such high precision. Pseudorange refers to the difference between the satellite clock time and the user clock time when the user clock time is always relatively imprecise [4]. The Achilles heel of GNSS is the inaccuracy of the receiver. The fourth satellite in any position fix is required to determine how far off precisely the receiver clock is compared with the satellite clocks. To underline how important this is, if the receiver clock is off by 1.25 s ($\approx 300\,000 \times 1.25$), the position would be in inaccurate by 375 000 km—the same distance as the moon's orbit from Earth.

In equation (3.2) [4], the pseudorange is expressed as

$$\rho_i = |r_i - r_u| + c \times b_u + \varepsilon_{pi}. \tag{3.2}$$

where $r_i$ is the satellite position at transmit time; $r_u$ is the receiver position at receive time; $b_u$ is the receiver clock bias expressed in seconds and $\varepsilon_{pi}$ the combined calculation for all the estimated or measured ionosphere and troposphere delays, clock mis-modelling, ephemeris and multipath.

## 4. Minimizing GNSS error with optimum satellite geometry

It is well documented that any GNSS requires at least four satellites for a full position fix and time or altitude fix. From the leading authority on space mission analysis and design, it is clear that the task of maximizing the

precision of a satellite constellation is a daunting one requiring highly detailed budget, equipment and orbit calculations [5]. However, three common denominator components are critical to minimizing errors created because of the geometry of satellites. These are the spacing, altitude and attitude of the satellites.

The quality of spacing of satellites has been termed as the geometric dilution of precision (GDOP). The errors experienced by GPS users at high latitudes can significantly depend upon GDOP. Equation (4.1) expresses how GDOP performance measurement is calculated, based on the variables of three dimensions plus time [4].

$$\text{GDOP} = \frac{1}{\sigma}\sqrt{\sigma_x^2 + \sigma_y^2 + \sigma_z^2 + \sigma_b^2}. \tag{4.1}$$

As a general rule, the wider the spacing of satellites the higher the quality of signal consistency. The lower the GDOP, the better the accuracy. By contrast, a constellation of satellites has a poor geometry when they are close together. Therefore, the GDOP is higher [6].

For any scientist operating in high latitudes or cryospheric regions, the task of determining GDOP before drone use may make sense because the drone can then be switched to the GNSS with the lowest GDOP. A GDOP between 2 and 5 is good and a GDOP of 1 is excellent. A website exists that allows this calculation to be made (www.calsky.com/cs.cgi/satellites (accessed 5 February 2020)). For example, on 13 July 2019 at 12.00 at Ny-Ålesund, for the six GLONASS satellites in view, the GDOP is 4.95. The GDOP for both GALILEO and GPS exceeded 5. On that date, any scientist operating a drone was best served by GLONASS.

# 5. Polar GNSS

It is insufficient to explain GNSS without a comparison of the leading systems in the context of high latitudes. No scientist needs to rely solely on GPS. Appendix A summarizes GNSS relevant to not only polar scientists, but also a regional satellite system, namely India's IRNSS relevant to cryospheric scientists researching in the Himalayas.

Figure 1 [7] records the ground tracks of the four main GNSS recorded in September 2013. Top to bottom, these are BEIDOU, GALILEO, GLONASS and GPS. What is apparent from all four is that high latitudes are less well served compared with equatorial regions. The BEIDOU tracks in the top graph show that these are confined to latitudes 55° N and 55° S. The third graph shows that GLONASS is the most likely to best serve scientists operating in polar regions.

Loss of GNSS signals is most evident when analysing how GPS performs at high latitudes compared with the Russian GLONASS system and the EU Galileo system. At first glance, these three systems seem similar. Currently, GPS functions with 31 operational satellites, GLONASS 24 and Galileo 24 [8]. The flight altitude of each are GPS at 20 180 km, GLONASS at 19 100 km and GALILEO at 23 222 km. So, each individual satellite in each constellation has a similar fraction of the Earth in view [4]. Thirdly, the attitude, by which is meant the satellite orbital inclination angle of each system are GPS at 55°, GLONASS at 64.8° and GALILEO at 56°.

The differences between these three constellations are clearly of importance based on these variables alone. With reference to GALILEO, it has been observed that factoring in its different 'orbital inclination and the flight altitude of the satellites will considerably increase the coverage of the polar regions, not so well achieved by GPS' [9]. Further, although GLONASS orbits at a similar altitude to GPS, its higher altitude orbit coverage over different higher latitudes is superior, as shown in table 1 [10].

From table 1, it can been deduced that GPS is not performing well at a ≥50° angle from 75° latitude. At the 75–80° latitude range, the percentage of GPS satellites is nearly half of the number of GLONASS satellites. Further, applying the minimum four satellite principle, at the 75–80° latitude range, GLONASS would still achieve a full position fix with five visible satellites, but GPS would provide just three satellites. Therefore, taking just two important high latitude location examples, a scientist on Svalbard (78° N) or Ellesmere Island (76° N) would be better served by ensuring that GLONASS is included as part of their navigation and positioning choices. In summary, in combination, these two GNSSs have played and logically will continue to provide vital navigation and positioning data at the poles.

# 6. Frame sample methodology

The fundamental aim of the drone survey was to target respondents that would meet the three quality survey implementation criteria of (i) reliability, (ii) validity, and (iii) representativeness [11,12]. The criterion of representativeness was the most challenging of these three, because it involved attempting to

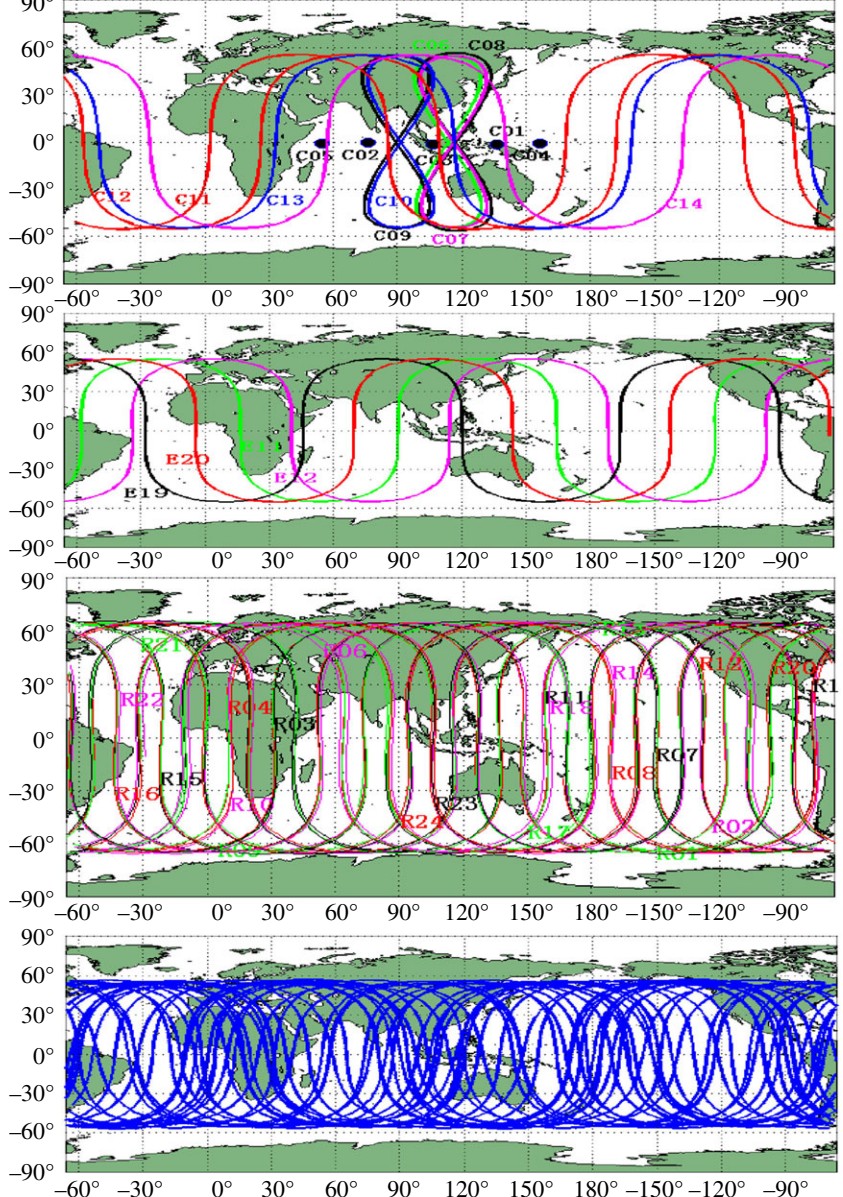

**Figure 1.** Four GNSS ground tracks.

**Table 1.** Percentage of visible satellites above angle H degrees.

| latitude ranges | SYS | 40°H | 50°H | 60°H |
| --- | --- | --- | --- | --- |
| 70–75° | GLONASS | 34.4 | 20.2 | 11.7 |
|  | GPS | 27.1 | 16.1 | 4.4 |
| 75–80° | GLONASS | 37.2 | **20.9** | 9.2 |
|  | GPS | 26.0 | **11.0** | 1.1 |
| 80–85° | GLONASS | 40.2 | 19.9 | 19.9 |
|  | GPS | 24.8 | 4.5 | 4.5 |
| 85–90° | GLONASS | 26.6 | 1.3 | 1.3 |
|  | GPS | 21.8 | 0.4 | 0.4 |

avoid over-concentration on clusters of researchers, for example just geomorphologists or only Arctic researchers. To maximize representativeness, five specialisms were finally selected: (a) Climate & Climate Change; (b) Ecology & Biodiversity; (c) Geomorphology; (d) Ice & Ice Movement; and (e) Oceanography.

**Table 2.** Drone survey summary.

| country of institute | no. researchers sent survey | no. researchers replying |
|---|---|---|
| Argentina | 6 | 0 |
| Australia | 9 | 1 |
| Brazil | 5 | 1 |
| Canada | 9 | 1 |
| Chile | 7 | 1 |
| China | 7 | 1 |
| Denmark | 7 | 0 |
| Finland | 12 | 3 |
| France | 12 | 4 |
| Germany | 23 | 5 |
| Italy | 8 | 0 |
| Japan | 5 | 0 |
| New Zealand | 12 | 2 |
| Norway | 13 | 3 |
| Russia | 11 | 1 |
| South Korea | 5 | 1 |
| Sweden | 11 | 5 |
| Switzerland | 11 | 4 |
| UK | 20 | 5 |
| USA | 18 | 4 |
| Total | 211 | 42 |

# 7. Target population of scientists

With the framework of the above five research specialisms (a–e), target scientists were selected from the leading polar and cryospheric institutions in Argentina, Australia, Brazil, Canada, Chile, China, Denmark, Finland, France, Germany, Italy, Japan, New Zealand, Norway, Russia, South Korea, Sweden, Switzerland, UK and USA. Added to this were the national polar research institute or institutes for each of these countries.

# 8. Choice of online survey platform

*Qualtrics*, a cloud-based survey solution was chosen (www.qualtrics.com (accessed 5 February 2020)). The *Qualtrics* software enabled a professional online questionnaire template to be developed. This extra effort meant that respondent scientists could answer survey questions in the minimum time with the least effort.

In early March 2019, an initial preparatory exercise involved sending 144 emails to polar and cryospheric researchers to understand who would be relevant inclusions in the final sampling frame of researchers using drones. After this exercise, the final number of potential respondents emailed was 211. Table 2 shows the subsequent 42 responses from both the Northern and Southern Hemispheres.

# 9. The drone survey questions

The polar scientists were asked to confirm the research areas in which they use drones. Based on 69 counts from the 42 respondents, figure 2 shows that Ice & Ice Movement had 18 drone users (26.09%), followed by Climate & Climate Change with 16 drone users (23.19%). A total of eight users (11.59%) recorded that their drones were used for Oceanography.

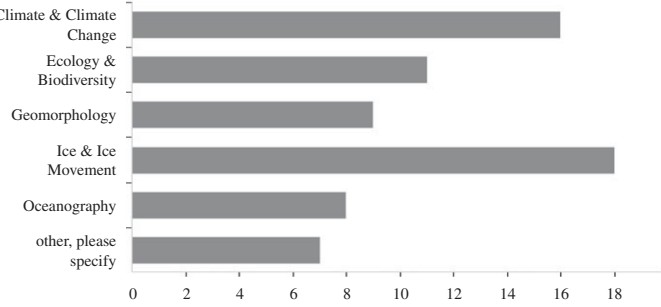

**Figure 2.** Research areas.

**Table 3.** GNSS choices by 33 aerial-only.

| sensor type | percentage |
| --- | --- |
| BEIDOU | 5.88 |
| GALILEO | 14.71 |
| GLONASS | 27.94 |
| GPS | 45.59 |
| IRNSS | 1.47 |
| other, please specify | 1.47 |
| don't know | 2.94 |
| total | 100 |

# 10. How much reliance was there on GPS?

For the 33 scientists using only aerial drones, the breakdown of GNSS choices is shown in table 3. There were nine scientists using both aerial and underwater drones, but their data were excluded because it proved to be unreliable.[1]

The scientists were asked the following question and options:

*When using your aerial drone for research applications, which of the following GNSS do you use for its navigation and positioning?*

— BEIDOU
— GALILEO
— GLONASS
— GPS
— IRNSS
— Other, please specify
— Don't know

An important aspect of this survey was the aim of establishing if there is an over-reliance by scientists on GPS (45.59%). At first glance, the 45.59% using GPS underlines that GPS is certainly important. However, for an inference to be drawn that there is an over-reliance on GPS, it was important to know what percentage of aerial-only GPS users were using GPS and GLONASS; or GPS, GLONASS and GALILEO.

Analysis of the 33 scientists using only aerial drones showed that 12 (36.36%) relied solely on GPS. Nine relied on GPS and GLONASS (27.27%). Another nine (27.27%) relied on GPS, GLONASS and GALILEO. Only one scientist used GPS and GALILEO, but not GLONASS. By contrast, when the 3.03% of scientists who did not answer the GNSS survey question are factored out, 60.61% of scientists use more than one GNSS.

[1]The nine scientists responded that they used both aerial and underwater drones. Unfortunately, their aerial drone data were unreliable through no fault of their own. By asking these nine scientists double the number of questions, so both aerial and underwater questions, both sets of answers were rushed. That is why 33 of the 42 respondents form the main sample analysed.

If the seven scientists that used both aerial and underwater are added to these results, the answer is not more insightful on GPS reliance percentages. With the scientists using both types of drone, the sample size is 40. Of these seven scientists, two (27.57%) relied solely on GPS. However, because five of the seven (71.42%) provided a 'Don't know' answer, this makes the data provided by the seven scientists using both aerial and underwater drones unreliable.

In summary, the 33 aerial-only scientists showed no over-reliance on GPS. The drone survey data on GNSS use for aerial-only scientists appear to be reliable, because all but one of the 33 scientists (96.96%) answered the GNSS use question with specific choices. One exception answered 'Don't know'.

## 11. GNSS combinations

Other results included the choice of GPS with GLONASS (27.27%). This is consistent with their established orbital operations for more than two decades over both polar regions.

Another aspect that points to increasing reliance on a trio of GNSS at high latitudes was the significant use of GALILEO (14.71%). This choice is consistent with engineering and mathematical analysis that the different 'orbital inclination and the flight altitude of the [GALILEO] satellites will considerably increase the coverage of the polar regions, not so well achieved by GPS' [8].

Furthermore, the drone survey revealed low-level use of both BEIDOU and IRNSS, which is consistent with the fact they both commenced operations as regional satellite systems rather than GNSS. Considerable expansion of BEIDOU means that it is now a potentially valuable global satellite system [13]. However, it was used by only 5.88% of the polar scientists.

## 12. Satellite-based augmentation system use

A satellite-based augmentation system (SBAS) is designed for all flight phases of commercial aircraft navigation. The SBAS provides a scientifically proven gap-filling service to mitigate GNSS errors. In each geographical region of Earth, the relevant SBAS provides the accuracy, availability and integrity needed to rely on a GNSS. This is achieved by providing, via a separate constellation of geostationary satellite signals, a set of positioning and time to the user's GNSS receiver. There are SBAS that cover GPS in North and South America including Southern Patagonia (WAAS and WAAS expansion), GALILEO in Europe including Iceland and Svalbard (EGNOS), and GLONASS across all Russia (SDCM) [14].

The drone survey showed that a high percentage of scientists (38.89%) made no use of any SBAS. Furthermore, an even higher percentage (41.67%) did not know what, if any, SBAS was used. The remaining respondents either used WAAS (13.89%) or EGNOS (5.56%). In summary, for this sample, the use of any SBAS was an insignificant part of scientific missions, and many scientists lacked awareness on whether any SBAS is relied on at all. One explanation for the low-level use is that at high latitudes, geostationary satellites have poor visibility [15].

## 13. Drone precision measurement choices and plans

Scientists were asked for information about (i) their current precise measurement choices and (ii) future planned measurement choices. As shown in table 4, the current and planned percentage changes are insignificant.

An exception was the planned use of the NASA GDGPS service. Subsequent to the drone survey completed in April 2019, on 7 May 2019, an email was sent to the NASA Jet Propulsion Laboratory (JPL) to understand if there was any reason known to JPL for this increased interest among polar users for GDGPS. The JPL technical manager replied that there was no specific reason or reasons known to them.

Having checked all the individual questionnaires, none of the current users intended to stop using GDGPS, so the real planned change (8.78%) indicated a significant increase in its use.

## 14. Ground control points

Ground control points (GCPs) have always played an important augmenting role to GNSS across many polar research areas. The mean number of GCPs used by the 33 polar scientists was 5.61, with a standard deviation of 3.29. The use of GCPs shown in table 5 indicates that 20 scientists (60.60%) currently use ≥1 GCPs.

**Table 4.** Precision measurement.

| precision research method | current (%) | planned (%) |
|---|---|---|
| differential GNSS using own base station | 28.33 | 23.88 |
| geo-referenced ground control points | 26.67 | 23.88 |
| inertial navigation system | 11.67 | 10.45 |
| iridium satellite constellation | 1.67 | 2.99 |
| *NASA* global differential GPS (GDGPS) | 1.67 | *10.45* |
| post-processing positioning | 20.00 | 16.42 |
| other, please specify | 5.00 | 4.48 |
| don't know | 5.00 | 7.46 |
| total | 100 | 100 |

**Table 5.** GCPs used by the 33 scientist.

| no. GCPs used | no. scientists | no. scientists as % of sample | avg. accuracy (m) | range (m) |
|---|---|---|---|---|
| 0 | 13 | 39.39 | 8.06 | 0.10–100 |
| 1 | 4 | 12.12 | 2.33 | 1–5 |
| 3 | 2 | 9.09 | 0.10 | 0.10–0.10 |
| 4 | 1 | 3.03 | 0.40 | 0.40–0.40 |
| 5 | 2 | 15.15 | 0.05 | 0.0001–0.10 |
| >5 | 1 | 3.03 | 0.50 | 0.50–0.50 |
| >10 | 10 | 30.30 | 1.38 | 0.0005–10 |
| totals | 33 | 100 | 1.76 | 0.0001–100 |

# 15. Regression calculations

The calculations sought to establish if there was a functional relationship between the accuracy of measurements obtained (the dependent variable or the response variable) with the following individual and pairings of independent variables:

(i) differential GNSS;
(ii) ≥1 GCPs used;
(iii) PPP;
(iv) differential GNSS + ≥1 GCPs;
(v) differential GNSS + PPP; and
(vi) differential GNSS + ≥1 GCPs + PPP.

Of these six relationship results in table 6, only differential GNSS + ≥1 GCPs + PPP had a significant $p$-value (0.05). Generally, in both engineering and physics, a $p$-value needs to be $\leq 0.05$ for the $R$-value to be significant.

Of the six regression calculations, the $R$ results for (i) the weak relationship between accuracy obtained and the use of differential GNSS (0.20), and (ii) accuracy obtained and the use of ≥1 GCPs (0.16), were the most surprising. A weak relationship is generally considered to be in the range of 0.20–0.39. These two augmentation techniques are well-established methods of increasing measurement accuracy.

Furthermore, the pairings of (iv) differential GNSS with ≥1 GCPs and (v) differential GNSS with PPP would both logically expect to have yielded a stronger $R$. However, their respective results (0.23 and 0.23) were also both in the weak range.

The $R$-value for (vi) differential GNSS, ≥1 GCPs plus PPP was 0.25. Its $p$-value was 0.05. Therefore, this was a close result. Notably, GNSS, ≥1 GCPs plus PPP has only a marginally stronger relationship (0.25) when compared with differential GNNS + ≥1 GCPs (0.23) or differential GNSS + PPP (0.23).

**Table 6.** Regression results.

| accuracy relationship | R | adj $R^2$ | p-value | scientists |
|---|---|---|---|---|
| Diff GNSS | 0.20 | n.a. | 0.07 | 33 |
| ≥1 GCPs | 0.16 | n.a. | 0.09 | 33 |
| PPP | 0.11 | n.a. | 0.13 | 33 |
| Diff GNSS + ≥1 GCPs | 0.23 | −0.007 | 0.14 | 33 |
| Diff GNSS + PPP | 0.23 | −0.005 | 0.12 | 33 |
| Diff GNSS + ≥1 GCPs + PPP | 0.25 | −0.03 | 0.05 | 33 |

Based on a meaningful sample of 33 scientists and a p-value at the statistically significant level, this is still the most valuable result of the six regression calculations.

## 16. Survey results

Overall, the survey results illuminated the limitations of analysing the multifaceted nature of GNSS drone navigation and positioning augmentation.

The response to the survey by the 33 aerial-only scientists was a representative sample of the scientific population using aerial drones in polar and cryospheric regions. Scientists from all five research areas answered along with a range of institutes from both the Northern and Southern Hemispheres. In this context, the sample was comprehensive and unbiased.

The sample of 33 aerial-only scientists was relatively small; therefore, the results have to be interpreted cautiously. However, the current total global population of polar and cryospheric scientists is certainly less than 300. Whatever the precise total population, 33 scientists is an insightful sample.

The detailed answers of all 33 aerial-only respondents showed a complex, multifaceted list of methods used. Different combinations, some without differential GNSS or without the use of GCPs, provided measurements at the sub-metre level.

## 17. Conclusion and future research

The drone survey data revealed three important points. First, answers from all 33 aerial-only respondents revealed that multifaceted combinations of measurement methods are used to augment GNSS signals. It was not possible to establish a strong pattern from any specific combination of augmentation methods that correlate with sub-metre accuracy.

Second, there was no over-reliance on GPS. Out of the 33 scientists using only aerial drones, just over one-third (36.36%) used solely GPS. Nine used GPS and GLONASS (27.27%). A further nine (27.27%) relied on GPS, GLONASS and GALILEO. Therefore, the risk of over-reliance on GPS does not apply to aerial-only scientists operating in polar or cryospheric regions.

Furthermore, polar scientists are aware of the inexorable trend of new GNSS technology providing more than GPS. This trend is driven partly by the decreasing cost of positioning, velocity and timing (PVT) semiconductor components in both drones and ground station receivers that provide multi-GNSS options. Typically, such PVT components contain BEIDOU, GALILEO, GLONASS and GPS options, with the ability to concurrently always provide for at least two of these GNSS.

Third, there is an unexplained projected increase in the use of NASA's GDGPS. The drone survey revealed a marked projected increase in the use of GDGPS; from 1.67 to 10.45%. Subsequently, NASA JPL has offered no explanation for this significant projected increase. Therefore, it would be valuable for future drone research to look at how the NASA JPL GDGPS service improves research for polar and cryospheric scientists. Specifically, the whole situation in Antarctica needs to be assessed, because the GDGPS augmentation network is prevalent on the Antarctic Peninsula, but not elsewhere [16].

## 18. Further research

Subject to research funding contraints, some polar scientists may benefit from assessing cutting-edge, privately owned satellite systems for drone PVT. As just one example, in 2019, Iridium completed the

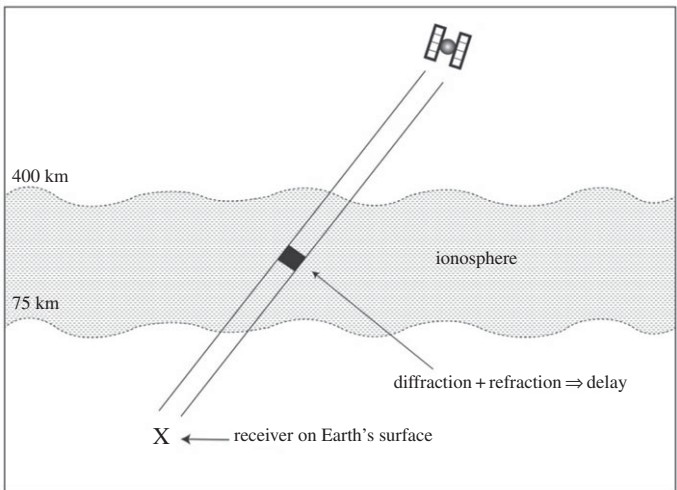

**Figure 3.** Ionospheric scintillation.

launching of its 66 low Earth orbiting (LEO) satellite system. This Iridium constellation carries Aireon aviation flight-tracking technology that allows commercial aircraft to transmit their GPS positions once every half second at any point around the planet. The Iridium satellites, as a network, transmit aircraft positions to a ground-level receiver. This Iridium constellation provides real-time, 100% coverage of Earth [17].

One consequence of the Aireon technology is an increased likelihood of locating a crashed aircraft. The permanent loss of aircraft, in the way that Malaysia Airlines Flight 370 disappeared in 2014, can be minimized. For high-value drones making beyond line of sight flights, this Aireon technology provides a higher probability of recovering crashed drones.

Second, there is the challenge of scintillation. At high latitudes, both the amplitude and phase of the received GNSS signal can be affected by scintillation. Amplitude scintillation occurs less frequently compared with phase scintillation [18]. At high latitudes, scintillation effects can last for many hours, even days [4]. Phase scintillation is important to polar scientists relying on the GNSS. Receiver measurements can result in '…positioning errors of tens of meters or, in the most severe cases, in complete outages due to Loss of Lock (LOL). Such a threat has a disruptive impact on sub-metre navigation and precise positioning' [19]. Figure 3 [4, adapted from diagram at p. 50] shows how scintillation effects GNSS signals.

Phase scintillation research reveals specific patterns that correlate with GNSS errors. For example, Pan & Yin [20] published an 'Analysis of polar ionospheric scintillation characteristics based on GPS data', derived from a 12-month, daily observation of scintillation in 2011 at the South Pole.

Their results revealed that phase scintillation followed a pattern of seasonal variation with two low periods starting in January and June. In April and October, it peaks [20]. For scientists researching with drones at high latitudes, these findings may point to the need to make most use of drones in the low-phase scintillation months. From a future research viewpoint, it would be valuable to know the effect of phase scintillation on drones using GNSS other than GPS, therefore focusing on BEIDOU, GALILEO and GLONASS as well.

Data accessibility. The data are deposited at Dryad Digital Repository: https://doi.org/10.5061/dryad.ht76hdrb1 [21].

Competing interests. I declare I have no competing interests.

Funding. I received no funding for this study.

Acknowledgements. The author completed this research as part of an MPhil in 2018–2019 at Cambridge University. I thank the following Scott Polar Research Institute (SPRI) members for their advice and insights: Gareth Rees, Tom Chudley, Neil Arnold, Peter Lund and Frances Marsh. Furthermore, thanks to Tom Jordan and Carl Robinson both of British Antarctic Survey (BAS), Dag Arne Lorentzen of the University Centre in Svalbard (UNIS), Tim Stockings (ex-BAS) and John Pottle of the Royal Institute of Navigation. I am also grateful to the anonymous reviewers who provided comments that improved the manuscript.

# Appendix A

Current and planned GNSS are given in table 7.

**Table 7.** Current and planned GNSS.

| system | accuracy and coverage | frequency | number and altitude | regime | additional facts |
|---|---|---|---|---|---|
| BEIDOU 北斗卫星导航系统 China www.cnsa.gov.cn | 10 m | 1.561098 GHz (B1) 1.589742 GHz (B1–2) 1.20714 GHz (B2) 1.26852 GHz (B3) | 35 orbiting at 21 150 km based on 2020 schedule | GEO(5) IGSO(3) MEO(27) | initially coverage was Long 70° E to 140° E; Lat 5° N to 55° N, so covers all Himalayas; 35 scheduled to orbit by 2020 |
| GALILEO EU www.gsa.europa.eu | 1 m global | 1.559–1.592 GHz (E1) 1.164–1.215 GHz (E5a/b) 1.260–1.300 GHz (E6) | 26 orbiting at 23 222 km | MEO(26) | hydrogen maser clock + reserve rubidium clock |
| GLONASS *ГЛОНАСС* *Russia* www.roscosmos.ru | 4.5–7.5 m global | 1.593–1.610 GHz (G1) 1.237–1.254 GHz (G2) 1.189–1.214 GHz (G3) | 27 orbiting at 19 130 km | MEO(27) | superior positioning where masking elevation >50° at latitudes >75° |
| GPS USA www.afspc.af.mil | 15 m global | 1.563–1.587 GHz (L1) 1.215–1.2396 GHz (L2) 1.164–1.189 GHz (L5) | 31 orbiting at 20 180 km | MEO(31) | continuously adapted with planned long-term investment |
| IRNSS (NAVIC) India www.isro.gov.in | 10 m regional, including all Himalayas | 1176.45 MHz (L5) 2492.028 MHz (S) | 7 orbiting at 36 000 km | high Earth (7) | Long 30° E to 130° E; Lat 30° S to 50° N, so covers all Himalayas |

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
