## [Reviewer comments · Royal Society Open Science]

Review History

RSOS-191494.R0 (Original submission)

Review form: Reviewer 1 (Katarina Gårdfeldt)

Is the manuscript scientifically sound in its present form?

Yes

Are the interpretations and conclusions justified by the results?

Yes

Is the language acceptable?

Yes

Do you have any ethical concerns with this paper?

No

Have you any concerns about statistical analyses in this paper?

No

Recommendation?

Accept as is

Comments to the Author(s)

The authors make a significant contribution to the field of using drones for earth observations, especially at high latitudes, by addressing the possible uncertainty when applying various positioning systems and drones.

The survey made in the study shows that among the responders GPS navigation system or GPS combined with other systems (e.g. GLONASS and GALILEO) were most common. One important contribution the authors make is to showcase the need of studying the effect of phase scintillation on various navigation systems over polar regions not only when using GPS but also e.g. BEIDOU, GALILEO and GLONASS systems for drone navigations.

Review form: Reviewer 2 (Joseph N. Pelton)

Is the manuscript scientifically sound in its present form?

Yes

Are the interpretations and conclusions justified by the results?

Yes

Is the language acceptable?

Yes

Do you have any ethical concerns with this paper?

No

Have you any concerns about statistical analyses in this paper?

Yes

Recommendation?

Accept with minor revision (please list in comments)

Comments to the Author(s)

This is a very interesting survey. The changing conditions in terms of the GNSS satellite systems, new capabilities such as the Aerion system on Iridium, and especially the changes in multi-system receivers and their declining costs should be noted.

Decision letter (RSOS-191494.R0)

24-Jan-2020

Dear Mr Sheridan,

On behalf of the Editors, I am pleased to inform you that your Manuscript RSOS-191494 entitled "Drones and Global Navigation Satellite Systems – current evidence from polar scientists" has been accepted for publication in Royal Society Open Science subject to minor revision in accordance with the referee suggestions. Please find the referees' comments at the end of this email.

The reviewers and handling editors have recommended publication, but also suggest some minor

revisions to your manuscript. Therefore, I invite you to respond to the comments and revise your manuscript.

- Ethics statement

- Data accessibility

If you wish to submit your supporting data or code to Dryad (<http://datadryad.org/>), or modify your current submission to dryad, please use the following link:
<http://datadryad.org/submit?journalID=RSOS&manu=RSOS-191494>

- Competing interests

- Authors' contributions

- Acknowledgements

- Funding statement

Please ensure you have prepared your revision in accordance with the guidance at <https://royalsociety.org/journals/authors/author-guidelines/> -- please note that we cannot publish your manuscript without the end statements. We have included a screenshot example of

the end statements for reference. If you feel that a given heading is not relevant to your paper, please nevertheless include the heading and explicitly state that it is not relevant to your work.

Because the schedule for publication is very tight, it is a condition of publication that you submit the revised version of your manuscript before 02-Feb-2020. Please note that the revision deadline will expire at 00.00am on this date. If you do not think you will be able to meet this date please let me know immediately.

If your manuscript is newly submitted and subsequently accepted for publication, you will be asked to pay the article processing charge, unless you request a waiver and this is approved by Royal Society Publishing. You can find out more about the charges at <https://royalsocietypublishing.org/rsos/charges>. Should you have any queries, please contact openscience@royalsociety.org.

on behalf of Dr Bethan Davies (Associate Editor) and R. Kerry Rowe (Subject Editor)
openscience@royalsociety.org

Reviewer comments to Author:

Reviewer: 1
Comments to the Author(s)

The authors make a significant contribution to the field of using drones for earth observations, especially at high latitudes, by addressing the possible uncertainty when applying various positioning systems and drones.

The survey made in the study shows that among the responders GPS navigation system or GPS combined with other systems (e.g. GLONASS and GALILEO) were most common. One important contribution the authors make is to showcase the need of studying the effect of phase scintillation on various navigation systems over polar regions not only when using GPS but also e.g. BEIDOU, GALILEO and GLONASS systems for drone navigations.

Reviewer: 2
Comments to the Author(s)

This is a very interesting survey. The changing conditions in terms of the GNSS satellite systems, new capabilities such as the Aerion system on Iridium, and especially the changes in multi-system receivers and their declining costs should be noted.

Author's Response to Decision Letter for (RSOS-191494.R0)

See Appendix A.

Decision letter (RSOS-191494.R1)

26-Feb-2020

Dear Mr Sheridan,

It is a pleasure to accept your manuscript entitled "Drones and Global Navigation Satellite Systems – current evidence from polar scientists" in its current form for publication in Royal Society Open Science.

on behalf of Dr Bethan Davies (Associate Editor) and R. Kerry Rowe (Subject Editor)
openscience@royalsociety.org

Appendix A

Reviewer 2 Comment 1 to the Author

The author should note the changing conditions in terms of GNSS satellite systems, such as the *Aerion* system on *Iridium*.

Author response added to article

Subject to research funding constraints, some polar scientists may benefit from assessing cutting-edge, privately-owned satellite systems for drone PVT. As just one example, in 2019 Iridium completed the launching of its 66 low Earth orbiting (LEO) satellite system. This Iridium constellation carries Aireon aviation flight tracking technology that allows commercial aircraft to transmit their GPS positions once every half second at any point around the planet. The Iridium satellites, as a network, transmit aircraft positions to a ground-level receiver. This Iridium constellation provides real-time, 100 per cent coverage of Earth.

One consequence of the Aireon technology is an increased likelihood of locating a crashed aircraft. The permanent loss of aircraft, in the way that Malaysia Airlines Flight 370 disappeared in 2014, can be minimised. For high value drones making beyond line of sight flights, this Aireon technology provides a higher probability of recovering crashed drones.

Reviewer 2 Comment 2 to the Author

The author should note the changing conditions in terms of GNSS satellite systems, especially the changes in multi-system receivers and their declining costs.

Author response added to article

Polar scientists are aware of the inexorable trend of new GNSS technology providing more than GPS. This trend is driven partly by the decreasing cost of positioning, velocity and timing (PVT) semiconductor components in both drones and ground station receivers that provide multi-GNSS options. Typically, such PVT components contain BEIDOU, GALILEO, GLONASS and GPS options, with the ability to concurrently always provide for at least two of these GNSS.